# Toxicological Assessment of Cross-Linked Beads of Chitosan-Alginate and *Aspergillus australensis* Biomass, with Efficiency as Biosorbent for Copper Removal

**DOI:** 10.3390/polym11020222

**Published:** 2019-01-30

**Authors:** Ana Gabriela Contreras-Cortés, Francisco Javier Almendariz-Tapia, Agustín Gómez-Álvarez, Armando Burgos-Hernández, Ana Guadalupe Luque-Alcaraz, Francisco Rodríguez-Félix, Manuel Ángel Quevedo-López, Maribel Plascencia-Jatomea

**Affiliations:** 1Microbiology and Micotoxins Laboratory, Departamento de Investigación y Posgrado en Alimentos, Universidad de Sonora, Hermosillo, Sonora, C.P. 83000, México; rayne2129@hotmail.com (A.G.C.-C.); armando.burgos@unison.mx (A.B.-H.); francisco.rodriguezfelix@unison.mx (F.R.-F.); 2Bioremediation Laboratory, Departamento de Ingeniería Química y Metalurgia, Universidad de Sonora, Hermosillo, Sonora, C.P. 83000, México; agustin.gomez@unison.mx; 3Universidad Estatal de Sonora, Hermosillo, Sonora 95370, México; luquealcarazana@gmail.com; 4Department of Materials Science & Engineering, University of Texas at Dallas, Richardson, TX 75080, USA; mquevedo@utdallas.edu

**Keywords:** cross-linking, biopolymer, fungi mycelium, composites, copper, biosorption

## Abstract

Sorbent materials of biological origin are considered as an alternative to the use of traditional methods in order to remove heavy metals. Interest in using these materials has increased over the past years due to their low cost and friendliness to the environment. The objective of this study was to synthesize and characterize cross-linked beads made of chitosan, alginate, and mycelium of a copper-tolerant strain of *Aspergillus australensis*. The acute toxicity of the biocomposite beads was assessed using brine shrimp *Artemia salina* nauplii and the phytotoxicity was determined using lettuce (*Lactuca sativa*) and chili pepper ’Anaheim’ (*Capsicum annuum*) seeds. The biosorption capacity for copper removal in simulated wastewater was also evaluated. Results showed that the biosorbent obtained had a maximal adsorption of 26.1 mg of Cu^2+^ per g of biocomposite, and removal efficiency was around 79%. The toxicity of simulated residual water after treatment with the biocomposite showed low toxicity toward seeds, which was highly dependent on the residual copper concentration. The toxicity of the biocomposite beads to *A. salina* was considered medium depending on the amount of the biocomposite, which was attributed to low pH. Biocomposite shows promise as biosorbent for the removal process of heavy metals.

## 1. Introduction

In recent years, there has been increasing concern for heavy metal pollution in superficial and groundwater due to the non-biodegradable characteristics of heavy metals and to their acute toxicity toward public health and the environment. The presence of heavy metals in wastewater is generated by anthropogenic activities such as mining, agriculture, the paper industry, among others [1]. Copper is an important and highly used metal due to its technological applications. It is an essential metal that participates in different key enzymes in all forms of life; however, high levels irritate the respiratory tract, and ingestion may cause nausea, vomiting, and diarrhea. An excess of copper in the blood could damage the liver and kidneys [2]. Epidemiologic reports show evidence that there is a high incidence of cancer among coppersmiths, and copper is suggested as the carcinogenic. An elevated incidence of stomach cancer in humans has been detected in areas with high levels of Zn:Cu in the soil [3].

The use of conventional methods for heavy metal removal, such as chemical precipitation, ionic exchange, and membrane filtration, among others, presents some disadvantages, such as a high cost or inefficient heavy metal removal when working with trace concentrations [4]. Currently, materials of biological origin are considered as an alternative to the use of traditional methods for removing these contaminants. Adsorbents such as clays [5], cellulose, alginate, chitin, and chitosan, are of interest due to their low cost, availability, and biosorption capacity. Some of these materials can be used in the form of hydrogels or hydrophilic gels, which are viscoelastic materials consisting of three-dimensionally cross-linked polymers that exhibit the ability to absorb a large quantity of water in a mechanically stable hydrophilic network, without dissolving and allowing free diffusion. These complex networks of cross-linked polymer chains form a unique group of materials, while being characterized with both solid and liquid-like properties. The cross-links vary among covalent, ionic, hydrogen, or those strongly hydrophobic nature [6]. Hydrogel beads made of polymers such as chitosan and alginate have been utilized as sorbent materials for heavy metal removal from wastewater.

Chitosan is a natural amino-polysaccharide product deriving from the deacetylation of chitin. Chitosan is widely studied for heavy metal removal due to its low cost, low or no toxicity, but mostly because of the number of hydroxyl and amino groups that are present, the latter conferring on chitosan a great number of active sites that are used to chelate heavy metals [7]. Notwithstanding this, one of the inconveniences of working with chitosan is its low mechanical strength and its low solubility in acidic media, which limit its function for wastewater treatment [8]. A chemical modification of chitosan through cross-linking comprises an efficient way to improve its chemical stability at a low pH [9]. Combining chitosan with other biopolymers such as cellulose, polylactic acid, and alginate is a strategy that can reinforce the biopolymer matrix and increase the functionality of the material. Alginate is a water-soluble polysaccharide that is extracted from brown seaweed. It is considered biodegradable and biocompatible and is characterized by its ability to form gels that can be induced by calcium ions [10]. It has been employed to encapsulate biomass, bacteria, and industrial waste [11,12,13]. Additionally, it is considered to have potential for heavy metal removal as well. When mixing alginate and chitosan, a polyelectrolyte complex (PEC) is formed through ionic interactions among the oppositely charged groups, typically constituted in an unordered polymeric network. The formation of an alginate-chitosan PEC is commonly achieved within pH ranges of 3.5–6.5, where both polymers are soluble [14].

Fungal mycelium is another biomaterial that can be combined with sorbent materials to increase heavy metal removal. Fungal biomass is abundant in the environment, can also be found in contaminated sites, can be easily and rapidly produced with a high yield using fermentation process at a relatively low cost, and can be easily manageable once obtained [15]. Fungal biomass possesses abundant functional groups that have been reported as responsible for heavy metal biosorption in the wall of fungi, such as carboxyl, amine, hydroxyl phosphate, and sulfonate [16]. The biomass of metal-tolerant microorganisms could also contain compounds that potentially increase the uptake or biosorption of these contaminants. Even though the potential benefits of these materials, the toxicological aspects related to its application in vegetal or animal models, have been poorly explored.

The purpose of this research was to synthesize a stable chitosan-based biomaterial that can be used for heavy metal removal under different environmental conditions. The chitosan-based biocomposite was achieved by combining chitosan with inactivated (dead) biomass of the copper-tolerant *Aspergillus australensis* fungus, alginate, and this was precipitated in tripolyphosphate (TPP) as a cross-linking agent. The obtained beads were evaluated in batch cultures for copper removal using simulated wastewater at different pH and copper concentrations. For the first time, toxicological analysis was performed using animal and vegetal models in order to evaluate the potential toxic effect of the obtained material on *Artemia salina* nauplii, lettuce (*Lactuca sativa*) seeds, and ’Anaheim’ chili pepper (*Capsicum annuum*) seeds as well.

## 2. Materials and Methods

### 2.1. Chitosan Gel Preparation

Two g of chitosan of medium molecular weight (CMMW) (448877, Sigma-Aldrich, St. Louis, MO, USA) was dissolved by stirring using 100 mL of acetic acid (99.8% purity, J.T. Baker, Center Valley, PA, USA) at 5% (w·v^−1^) [17].

### 2.2. Alginate Gel Preparation

One g of sodium alginate (W201502, Sigma-Aldrich, St. Louis, MO, USA) was dissolved in 100 mL of hot (60–70 °C) distilled deionized water until its dissolution [18].

### 2.3. Production of Aspergillus australensis Biomass

Inactivated (or dead) biomass powder was prepared from growing spores of *A. australensis* in Czapek broth enriched with yeast extract. The broth had the following composition: K_2_HPO_4_ 1 g·L^−1^; NaNO_3_ 3 g·L^−1^; MgSO_4_ 0.5 g·L^−1^; KCl 0.5 g·L^−1^; FeSO_4_ 0.01 g·L^−1^; sucrose 30 g·L^−1^, and yeast extract 5 g·L^−1^; also, a known copper sulfate (CuSO_4_) concentration (50 mg·L^−1^) was added. One hundred mL of the broth was added to an Erlenmeyer flask, and the broth was later inoculated with a spore suspension of 5 × 10^5^ spores·mL^−1^. Flasks were incubated in an orbital shaker at 150 rpm at 30 ± 1 °C for 5 days. Then, the produced biomass was vacuum-filtered and washed with deionized distilled water until a pH of ≤ 8 [19]. For inactivation, biomass was sterilized by autocleavage for 20 min at 121 °C and dried at 60 °C for 24 h. The dried biomass was milled and homogenized using a mortar until a fine powder was obtained.

### 2.4. Synthesis of Cross-Linked Composites Beads

The synthesis of chitosan composite beads was accomplished in two phases [17,18]. Phase 1: 8.5 mL of CMMW (20 g·L^−1^), was placed in a beaker. To this solution, 0.05 g of dead fungal powder biomass was added and the solution was homogenized for 45 min. After that, 1.5 mL of sodium alginate solution at 1% (w·v^−1^) was added. The mixture was maintained under agitation for 2 h in order to obtain the polymerization gel. Phase 2: The polymerization gel obtained in Phase 1 was dripped using a peristaltic mini-pump (Fisher Scientific 13-876-2 variable-flow peristaltic pump, low flow, Pittsburgh, PA, USA) into a solution of TPP (Sigma-Aldrich, St. Louis, MO, USA) with a concentration of 1 g·L^−1^ in order to precipitate and form the composite beads. The size of the composite beads was determined using a graduated scale in cm and a stereomicroscope. Size was reported as the average of 10–15 measurements of the diameter of the beads individually.

### 2.5. Fourier-Transform Infrared Spectroscopy (FT-IR)

The FT-IR characterization of cross-linked chitosan with biomass, alginate, and TPP was obtained using a Thermo Scientific Nicolet iS50 FT-IR spectrometer (Madison, WI, USA). Measurements were performed at room temperature, and an average of 32 scans in a spectral range of 4000–400 cm^−1^ were recorded. The analysis was performed in liquid media (water) and the attenuated total reflection (ATR) accessory was used. Spectra of pure CMMW, alginate, and biomass were also obtained to elucidate the interactions.

### 2.6. Biosorption of Copper Using Composites Beads

#### 2.6.1. Biosorption of Copper Using Fresh Composites Beads

A total of 0.15 g of fresh composite beads were place in Erlenmeyer flasks with 150 mL of a Cu^2+^ solution containing 20 ppm of Cu^2+^. A solution with no Cu^2+^ was employed as control. Different conditions of pH were tested (5.5, 5.0, and 4.5, respectively), at 35 °C (temperature selected based on the high copper-adsorption of the fungal biomass) (data not shown). The range of pH was selected based on the copper solubility to avoid the metal precipitation, which could occur at pH > 6 and high copper concentration. Flasks were incubated in an orbital shaker at 120 rpm at 35 °C. Samples of 10 mL were taken under shaking and by assuring that the solid/liquid ratio were not changed by this operation. Samplings took place during different time periods (at 0, 15, 30, 45, 60, 80, and 100 min, and at 24 h). Furthermore, samples were preserved with HNO_3_ (J. T. Baker) and analyzed by atomic adsorption spectroscopy (AAS) using an AAnalyst^TM^ 400 AA spectrometer (PerkinElmer, Inc., Waltham, MA, USA).

#### 2.6.2. Biosorption of Copper Using Lyophilized Composites Beads

A total of 0.15 g of lyophilized composite beads was placed in Erlenmeyer flasks with 150 mL of a Cu^2+^ solution at different concentrations of copper (20, 50, 100, and 200 ppm of Cu^2+^). A solution with no Cu^2+^ and no composited beads was utilized as control; also, a treatment of pure chitosan beads was also evaluated for biosorption capacity to compare against the biosorption with cross-linked chitosan composite beads. For the experiment using 20 ppm of Cu^2+^, three pH conditions were evaluated (5.5, 5.0, and 4.5, respectively). For biosorption kinetics using 50, 100, and 200 ppm of Cu^2+^, the pH used was 5.0. Temperature, shaker, and the samples taken were under the same conditions as mentioned in Section 2.6.1.

### 2.7. Quantification of Copper and Biosorption Capacity of Composites Beads

The samples were analyzed by AAS using an air/acetylene flame. The amount of copper adsorbed by the composite beads was calculated using Equation 1 [20]:q = ((Ci − Cf)/m) V(1)
where: q = mg of metal ions adsorbed per g of biomass (mg·L^−1^), Ci = initial metal ion concentration (mg·L^−1^), Cf = final metal ion concentration (mg·L^−1^), m = dried biomass (g), and V = solution volume (L).

### 2.8. Langmuir Isotherm Studies

Equilibrium experiments for the isotherm studies were evaluated at concentrations of 20, 50, 100, and 200 ppm of Cu^2+^. The final concentrations were measured, and experimental data was adjusted to the Langmuir model (Equation 2):q_e_ = (q_max_ b Ce)/(1 + b Ce)(2)
where: q_e_ = adsorption capacity at equilibrium (mg·g^−1^), q_max_ = maximal adsorption capacity (mg·g^−1^), Ce = concentration of the ion at equilibrium (mg·L^−1^), and b = Langmuir constant. This model demonstrates a limit of adsorption that can be interpreted as a complete covering of the adsorbent surface by a maximal amount of adsorbate (monolayer limit) [21]. Langmuir isotherm model assumes the formation of an adsorbed solute monolayer on a uniform surface with a finite number of adsorption sites. This means that when a site is filled, no further sorption can take place at that site; therefore, the surface will reach a saturation point where the maximum adsorption of the surface will be achieved. In addition, the Langmuir constant (b) is related to the adsorption energy that may be interpreted as follows: b < 0: unfavorable adsorption; b = 0: linear adsorption with no specific affinity; b > 0: favorable adsorption. Higher values of b suggest better affinity.

### 2.9. Toxicological Assays

#### 2.9.1. Acute Toxicity on Brine Shrimp *Artemia salina* Nauplii

The acute toxicity test with *A. salina* nauplii was carried out based on the method of Molina-Salina and Said-Fernández [22]. In order to hatch *A. salina* eggs, 5 g of eggs was placed in Erlenmeyer flasks with sterile seawater. Aeration was maintained with an air pump from a fish tank, artificial illumination was turned on for 24 h, and the temperature was maintained at 25 °C. After hatching, the nauplii of *A. salina* were separated into groups of 10 specimens. Each group was placed in test tubes containing 5 mL of seawater and the substance to be tested (water from biosorption kinetics, 1 g of CMMW beads, 1 g of alginate beads, 0.25–1.0 g of CMMW-fungal biomass-alginate composite beads). Nauplii of *A. salina* with only seawater served as control. The test tubes were maintained under illumination for 24 h. After the incubation period, the surviving nauplii were counted with the aid of a magnifying glass and the percentage of survival was determined.

#### 2.9.2. Acute phytotoxicity on lettuce (*Lactuca sativa*) and chili pepper (*Capsicum annuum*) seeds

The lettuce seeds were obtained from a local store in Hermosillo, Sonora, Mexico, procuring that the selected seed were uncured (no pesticides), with good germinating power and low elongation variability. Filter papers were placed in Petri dishes and 4 mL of a residual solution of Cu^2+^ (from biosorption kinetics) was added to avoid the formation of air bags. Subsequently, 20 seeds were placed on the filter paper, leaving a space for root elongation between every seed. Petri dishes were sealed and placed in plastic bags to avoid the loss of humidity; additionally, the bag was covered with aluminum foil to incubate the seeds with no light for a period of 7 days at a temperature of 22–25 ± 1 °C. Three repetitions per treatment were carried out. Seeds with distilled deionized water served as control. The variables measured included percentage of germination and root elongation [23]. The same procedure was followed in order to test ’Anaheim’ chili pepper seeds, but the incubation period was increased to 14 days.

## 3. Results and Discussion

### 3.1. Synthesis of Composites Beads

The composite beads obtained had a homogenous average size of 0.2 ± 0.01 cm in fresh state; the beads were of an opaque white color and were a consistency firm to the touch. For lyophilized composite beads, the size was 0.1 ± 0.01 cm; they presented a pale beige color and also had a firm consistency, but were more fragile than fresh composite beads (Figure 1). The size obtained for composite beads was acceptable and could be used in different types of biosorption experiments such as batch, columns, or fixed beads. Fresh (Figure 1a) and lyophilized (Figure 1b) beads show a different size because lyophilized beads lose all water content resulting in a decrease in size [24]. Fresh beads have around 90%–95% of water content, which confirms the hydrophilic nature of the beads [25].

### 3.2. FT-IR Analysis of the Composite Beads

In Figure 2, the FT-IR spectra for the different biomaterials are presented. As it can be observed for Figure 2d, the spectra for CMMW present the region of 3372 cm^−1^, which corresponds to –OH and –NH groups, and vibrations and stretching of the –NH bond is observed. In the region of 3022 and 2753 cm^−1^, stretching of C–H occurs [26,27]. In the region of 1528 cm^−1^, there was stretching of C=O corresponding to amide I [28]. For the alginate spectrum (Figure 2c), the region of 3257 cm^−1^ corresponds to the stretching of –OH groups. In the region of the 1599 and 1405 cm^−1^, carboxylic groups and stretching of –COO^–^ are present. The peak at 1025 cm^−1^ depicts stretching of C–O–C [29].

In the spectrum of the fungal biomass (Figure 2b), characteristic groups were found in the region of 3298 cm^−1^, corresponding to the asymmetric stretching of NH_2_ of the amines and to the presence of –OH groups. The signal at 2922 cm^−1^ reveals stretching of the C–H groups. For the signal at 1645 cm^−1^, there is stretching of C=O due to the deformation of –NH, which indicates the presence of amide I. The signal in 1020 cm^−1^ indicates the stretching of C–O carboxylic groups. The signals reported in this spectrum are similar to those found in *Penicillium chrysogenum* biomass [30].

In Figure 2a, the spectra of the composite C+B+A is presented. As can be observed, there is stretching in the region between 2750 cm^−1^ and 3400, where –NH groups and –OH groups can be found. For the region of 622 and 1531 cm^−1^, stretching of C–O and –NH is found. The region of 1028 cm^−1^ corresponds to the stretching of C–O and C–C. Since chitosan is the major component of the composite, the majority of the interactions occur around it. TPP acts as a cross-linking agent between the biopolymers and the formation of the composite beads, which can be attributed to ionic gelation [31]. The charges of the –NH^3+^ groups of chitosan interact with the dissociated products of TPP in solution (P_3_O_10_^5−^ and HP_3_O_10_^4−^); this interaction was described by Goycoolea et al. [32] during the synthesis of the chitosan-alginate-blended nanoparticles employed for the delivery of macromolecules. At the same time, it is considered that the chitosan –NH^3+^ groups will interact with the negatively charged –COO^−^ of the alginate and carboxylic groups of the fungal biomass. This can be due to a significant decrease that was found in the signal between 1600 and 1500 cm^−1^, which can be attributed to the interaction of amino groups with carboxylic and carboxylate groups [33]. It is possible to observe a signal near 1650 cm^−1^, which can indicate interaction with the carbonyl groups present in the biomass and in the alginate with the amino groups of the chitosan chain [34]. Fungal biomass possesses a great amount of hexamines and proteins, and these are among the main constituents of the fungal cell wall [35]. By adding the fungal biomass to the mixture, it is provided with additional –NH, –OH, and carboxylic groups that can interact in a similar manner to that of TPP, as described previously.

### 3.3. Biosorption of Copper by Biocomposites Beads

In Figure 3, the results of the biosorption kinetics are presented for fresh and lyophilized composite beads. For fresh composite beads, the copper concentration decreased in all of the pH evaluated (5.5, 5.0, and 4.5) (Figure 3a). The highest metal removal was observed at pH 4.5, decreasing copper concentration to a value of 10.48 ± 0.54 ppm of Cu^2+^ compared to the control (21.67 ± 0.07 ppm of Cu^2+^). This represents a copper removal of 51.8% (Table 1). Furthermore, at pH 5.5 and pH 5.0, a removal of 43.6% and 47.0%, respectively, was also achieved.

The values obtained in the treatments are considered acceptable, because one half of the initial concentration utilized (20 ppm of Cu^2+^) was removed and could further be increased by 2-way kinetics. The adsorption capacity of the material was 11.19 mg of Cu^2+^ per g of composite.

On comparing these results with those of Yu et al. [36], in which these latter authors synthesized chitosan-TPP nanoparticles and alginate microparticles for copper removal (0.5–50 mM), it was found that the fresh particles could remove copper and that this was dependent on the concentration of chitosan during synthesis, with maximal adsorption of 7.93 mg·g^−1^ of particles [chitosan 2 mg·mL^−1^ (1.0 TPP)], 4.24 mg·g^−1^ of particles [2 mg·g^−1^ de particles (0.7) TPP], and 4.87 mg·g^−1^ of particles [chitosan 4 mg·mL^−1^ (1.0 TPP)] after 1 week of exposure to the metal, contrary to what was expected by increasing surface area, the low size of the particles (18–244 nm) reduced adsorption capacity. By combining chitosan and alginate into particles, copper removal was increased to 12.63 mg·g^−1^ of particles.

Sánchez-Duarte et al. [37] used pure chitosan beads (c-pure), and cross-linked with TPP (c-TPP) or with glutaraldehyde (q-Glu) to remove Cu^2+^ from an initial concentration of 416 ppm. The authors determined that 1 g of beads could remove 15.91%, 12.81%, and 7.76% of metal, respectively. Additionally, the authors also determined that by increasing the amount of beads, a higher removal of Cu^2+^ could be achieved (4.5 g of c-pure beads removed 99.35% in 4 h). Based on the latter, the results suggest that the fresh chitosan-alginate-fungal biomass composite synthesized in this study (at the size obtained and the amount used of material) revealed high potential for copper removal.

The results obtained with lyophilized composite beads are presented in Figure 3b. The copper concentration also decreased in the pH evaluated where, at pH 4.5, 5.0 and 5.5, copper removal was observed with values of 5.15 ± 0.15, 4.39 ± 0.19 and 4.72 ± 0.20 ppm of Cu^2+^, respectively. Removal efficiency was found to be 76.2%, 79.8% and 78.2% (Table 1), where no differences were found among the treatments. The amount of metal removed by lyophilized composite beads was around 17.11 mg of Cu^2+^ per g of composite. Comparing the results obtained with fresh composite beads, the lyophilized composite beads performed much better. This could be due to the lack of interference related to water molecules that are not present in the pores of the material, which made the latter a more efficient biosorbent.

Biosorption capacity for pure chitosan beads was also evaluated, which were capable of reducing Cu^2+^ concentration to a value of 1.58 ± 0.94 ppm compared to control (21.67 ± 0.07 ppm). This represents a 92.7% of metal removal. Pure chitosan beads are able to remove a higher amount of Cu^2+^ because some active sites are lost during cross-linking process. However, the low mechanical properties of pure chitosan beads after lyophilization are not adequate for biosorption process; these beads can easily break, loose shape, and generate particles with different sizes that are only removed by filtration or centrifugation in comparison to the cross-linked chitosan beads, which can be easily removed by decantation and are resistant to manipulation. In addition, it is reported that pure chitosan beads tend to protonate amino groups present in acidic pH, which causes a partial or total dissolution of pure chitosan in acidic solutions [37]. The cross-linked chitosan beads in this study were resistant to acidic conditions.

The effect of the pH can be observed during the biosorption process, and it was determined that pH 5.0 is better for copper removal. This results are similar to those of Giraldo et al. [38], who reported that the removal capacity of chitosan-TPP beads was reduced in a more acidic pH. This was attributed to the amine groups of the beads that began to protonate, causing electrostatic repulsion between protons and cations in adsorption sites. Biosorption of metals is highly dependent on pH conditions. Cations and protons compete for binding sites, which means that the biosorption of metals such as Cu, Cd, Ni, Co, and Zn is reduced at low pH values (below 3.0) [39]. In the present study, pH values within 4.5–5.5 ranges were used based on a previous study on biosorption processes, in which *Aspergillus australensis* biomass (data not shown) performed better for copper removal within this pH range. Additionally, at a pH above 5.8–6, a precipitation of the metal can occur in the form of Cu(OH)_2_ hydroxides, therefore it is not recommendable to carry out biosorption processes at alkaline conditions.

#### Effect of Copper Concentration

In order to determine the effect of the copper concentration during biosorption experiments, different copper concentrations were used (50, 100, and 200 ppm) (Figure 4).

These experiments were only performed with lyophilized composite beads since biosorption kinetics with 20 ppm of Cu^2+^ revealed that these were more efficient at removing copper compared to fresh composite beads. According to the results obtained, from an initial concentration of 50 ppm of Cu^2+^, the metal concentration was reduced to a value of 25.6 ± 0.40 ppm of Cu^2+^ after the composites were added (Figure 4a).

In Figure 4b, for the concentration of 100 ppm of Cu^2+^, the concentration was reduced to a final value of 66.39 ± 0.87 ppm, and for 200 ppm of Cu^2+^, the concentration was reduced to 151 ± 0.74 ppm. This represents a removal efficiency of 52%, 39.3%, and 23.2%, respectively (Table 2). With the results obtained in this kinetic, the Langmuir model was applied (Figure 5, Table 2) and analysis showed that the amount of copper that the composite beads could adsorb was higher with increasing copper concentration. Notwithstanding a decrease in removal efficiency (%) was observed for biosorption process, but this could be due to the increase in Cu^2+^ concentration; nonetheless, a higher amount of metal was adsorbed for each concentration evaluated.

The value of q_max_ was 26.56 mg Cu^2+^ g^−1^ of the composite beads, b (the Langmuir constant) was 0.358, and R^2^ (the correlation coeficient) had a value of 0.927. The values obtained indicate that the Langmuir model adjusts to the experimental data obtained and are adequate to describe the adsorption process in this study. These results suggest that biosorption equilibrium mostly occurs during the first 100 min after the contact of the composite with simulated wastewater. These results are similiar to those reported by Rosaria Panuccio et al. [40] and Ngah et al. [4], in which the equilibrium was reached at at approximately the same time. The composite beads presented in this research achieved a maximal adsorption capacity of 26.1 mg of Cu^2+^ g^−1^ of composite beads, which is similar to that obtained by Ngha et al. [4], who used chitosan-TPP beads to remove a maximal value of 26.06 mg of Cu^2+^ g^-1^ of composite bead. Tsai et al. [41] made chitosan-coated montmorillonite beads to remove copper and achieved a maximum biosorption of 13.04 mg of Cu^2+^ per g of composite bead. Comparing with other adsorbent materials, Mokhter et al. [42] used modified silica-chitosan to remove 68 mg·g^−1^ of Cu^2+^. Shim et al. [43] and Monser and Adhoum [44] used activated carbon and they achieved to removed 9–38 mg·g^−1^ copper; Babel et al. [45] used clay adsorbents to remove around 1–25 mg·g^−1^ copper as well. Comparing the values of copper removal reported in the literature, it is considered that the composites of chitosan cross-linked with fungal biomass, alginate, and TPP have a good biosorption capacity for metals such as copper, and they represent a feasible and low-cost option for biosorbent process application.

### 3.4. Toxicological Assays

#### 3.4.1. Acute Toxicity on Brine Shrimp *Artemia salina*

After the exposure of brine shrimp *Artemia salina* to the different materials, it was observed that pure CMMW beads, alginate beads, and dead fungal biomass had no toxic effect on *A. salina*, based on the percentage of survival, which was 90%, 96%, and 100%, respectively. These results were similar to those of the control (100% survival) (Figure 6). For the cross-linked composite C+B+A treatment in proportions of 1.0, 0.75, 0.5, and 0.25 g, survival of the brine shrimp was reduced, with percentages of survival of 58%, 56%, 44%, and 10%, respectively. This can be attributed to the presence of the acetic acid necessary for chitosan composite gel preparation, which lowers the pH of the solution to values of 4.5–5.5 depending on the amount of composite used, which could explain the low survival of brine shrimp.

*A. salina* requires a pH of 7.0–8.5 for optimal survival [46,47]. In the case of lyophilized composite beads, no survival was observed. This can be attributed to the fact that lyophilized composite beads remain at the surface of the water, which could have reduced the amount of oxygen necessary for their survival [47]. As mentioned previously, the survival of brine shrimp in the presence of dead fungal biomass was 100%, indicating that no toxic materials were present. Microbial biomass contains several components, such as proteins, carbohydrates, among others, that could be employed as a food source. Toi et al. [48] reported the use of bacterial biomass as a nutrient source for *Artemia* when limiting conditions of algae are present in the growth medium. *A. salina* is a non-selective filter feeder and can consume micro-algae, bacteria, protozoa, and detritus particles. Fernández [49] specified that food size for *Artemia* can range from between 6.8 and 27.5 μm, but adults are able to ingest all particles with a size below 50 μm. The biomass obtained in this study was pulverized into a fine dust for optimal incorporation into the composite mixture.

#### 3.4.2. Acute Phytotoxicity on *Lactuca sativa* Seeds

Seeds that were exposed to pure CMMW beads, alginate beads, and lyophilized composite beads demonstrated a decrease in the germination index of 68%, 61%, and 48%, respectively, compared with that of the control (87%) (Figure 7). For pure chitosan beads, this can be attributed to the pH of the beads, which is around 12. Beads were rinsed with aboundant water, however alkalinity remained after precipitation in NaOH.

Lettuce seeds require a pH between 6.5 and 7.0 in order to develop fully [50]. High alkalinity in a pure chitosan treatment can give rise to low absorption of nutrients necessary for seed germination and growth [51]. This could explain why root growth was affected in CMMW beads, with a value of 0.14 ± 0.03 cm in comparison with the control (0.48 ± 0.24 cm). Despite alginate having a lower germination index compared to that of the control, root develepmont revealed a value of 0.45 ± 0.06 cm, which was similar to the control. The lower germination in the alginate treatment could be due to variations in the handling and quality of the biological material employed in the treatment [52] and not to a toxic effect, because alginate is considered non-toxic and is widely utilized in different immobilization processes for plant and vegetable tissue. However, it is considered that development in this media can be a slower process [53,54].

For lyophilized composite beads, it is possible that water competition occurred, since composited beads are completely dry. Additionally, as observed in previous shrimp-brine studies (Section 3.4.1), the respiration process could have been diminished by the lyophilized beads on reducing seed germination and root growth. Root elongation also was lower, 0.10 ± 0.05 cm, compared with that of the control (0.48 ± 0.24 cm), and can be attributed to the lack of sufficient water.

In the case of the fresh cross-linked composite beads with biomass, alginate, and TPP, the germination index was 82.5%, similar to the germination of the control. This could be attributed to the presence of the biomass of *Aspergillus australensis*. Microbial biomass possesses different components, among these a great amount of proteins present in the cell wall that could provide additional nutrients for adequate seed germination.

Root elongation was lower compared to that of the control, with values of 0.17 ± 0.05 and 0.48 ± 0.24 cm, respectively, which can be attributed to the pH of the fresh beads (4.5). By having a low pH during the seed germination of the lettuce seeds, a decrease in root elongation can be found, as previously described in the study by Inoue et al. [55]. These authors mention that root elongation is decreased when the seeds are exposed to a pH below 5, and that this can completely inhibit root elongation at a pH of 3.5.

In Figure 7b, the effect of the residual water from biosorption kinetics are presented and, as it can be observed, no statistical differences (p > 0.05) were found among the treatments, indicating that the different waters after treatment are potentially equally harmful for seed germination. The treatment that was less affected was the residual water of the kinetics with 20 ppm of Cu^2+^ and pH 5.5, were seed germination was similar to control. Root elongation did not appear to be affected, with values ranging from 0.25 ± 0.08 to 0.7 ± 0.2 cm for the majority of the treatments. The only treatment that exhibited a reduced root elongation was that of the residual water used from 100 ppm of Cu^2+^ kinetics. This can be attributed to the high copper concentration to which the roots were exposed and by which they were subsequently affected. Copper can affect the biosorption of nutrients and other important physiological processes when present in excess, for example, at concentrations of 20–30 µg·g^−1^ dry weight [56,57].

#### 3.4.3. Acute Phytotoxicity on Chili Pepper (*Capsicum annuum*) Seeds

Treatment with alginate beads did not inhibit chili-pepper seed germination and was similar to the control (Figure 8a). On the other hand, pure CMMW beads showed a decrease in seed germination, which can also be attributed to the pH of chitosan being extremely alkaline after bead formation, affecting seed germination [51] and root elongation (0.10 ± 0 cm) compared to the control (2.06 ± 0.4 cm), as noted in Section 3.4.2. Fresh cross-linked composite beads also affected seed germination (53 %) and root elongation 0.43 ± 0.03 cm compared to those of the control (2.06 ± 0.4 cm). The low root elongation can be attributed to the low pH, reducing root elongation (pH 4.5), which also affected nutrient adsorption, as mentioned in Section 3.4.2.

Chili pepper seeds germinate at a pH of 6.8–7.5 [58], which could explain the low germination in the seeds. In the case of the cross-linked composite bead, the presence of the biomass does not appear to exert an effect on seed germination compared to the assay with lettuce seeds, and it is considered that chili pepper seeds were those most affected by the pH of the composite beads. However, root elongation was less affected (0.43 ± 0.03 cm) compared with the treatment with pure CMMW beads.

In Figure 8b, the germination of chili pepper seeds exposed to the residual-simulated wastewater from biosorption kinetics is presented. No effects were observed during seed germination among the different treatments, with a germination index between 83% and 90%, and no statistical differences (p > 0.05) were found when compared to the control. This indicates that there were no harmful residues after the biosorption experiments in the remaining water after the treatment with composite beads. However, a decrease in root elongation in treatments with the residual water from biosorption kinetics (50 and 100 ppm) was observed, with a root elongation of 0.36 ± 0.02 to 0.39 ± 0.09 cm, respectively, compared to that of the control (2.06 ± 0.4 cm). This can be attributed to the presence of copper, as noted in Section 3.4.2, in that residual concentrations of the metal can still be found [56].

In this study copper concentration was reduced to a minimum of 4.72 ± 0.2–4.39 ± 0.19 ppm of Cu^2+^. As previously mentioned, the remaining concentration was not below toxic levels for seed germination and adequate growth of lettuce and chili pepper seeds. However, copper concentration can further be decreased by implementing two-way kinetics strategies to increase copper removal; therefore, more studies are needed to optimize the biosorption process presented in this study.

## 4. Conclusions

Adsorption experiments showed that cross-linked chitosan with biomass-alginate and TPP composite beads can be used as an environmentally safe and effective biosorbent for copper removal. The Langmuir model demonstrated that the maximal adsorption capacity was 26.56 mg Cu^2+^ per g of composite beads. An increased copper removal can be achieved at pH 5.0 and 35 °C; however, experimental data showed that composite beads can perform well in other pH. Lyophilized composite beads had a better removal capacity compared to that of fresh composite beads, and it is considered that lyophilized composite beads would perform better in biosorption processes using column systems or bioreactors due to their high adsorption capacity compared to fresh composites. The cross-linked chitosan composites exhibited toxicity to *A. salina*, which was attributed to acetic acid that lowers pH below the recommended levels for the survival of nauplii. Toxicological evaluation revealed that water after copper removal could be potentially harmful to the seed germination process of lettuce and chili pepper seeds; however the toxicity is related to the copper concentration and not the material itself. It is important to ensure that the copper concentration is below toxic levels for plants. Further studies are suggested to ensure the full safety of the material to be used in the environment.

## Figures and Tables

**Figure 1 polymers-11-00222-f001:**
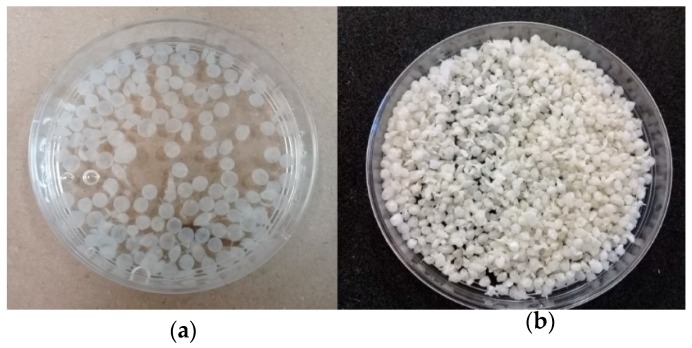
Physical appearance of synthesized cross-linked chitosan-based beads: (**a**) fresh composites, and (**b**) lyophilized composites.

**Figure 2 polymers-11-00222-f002:**
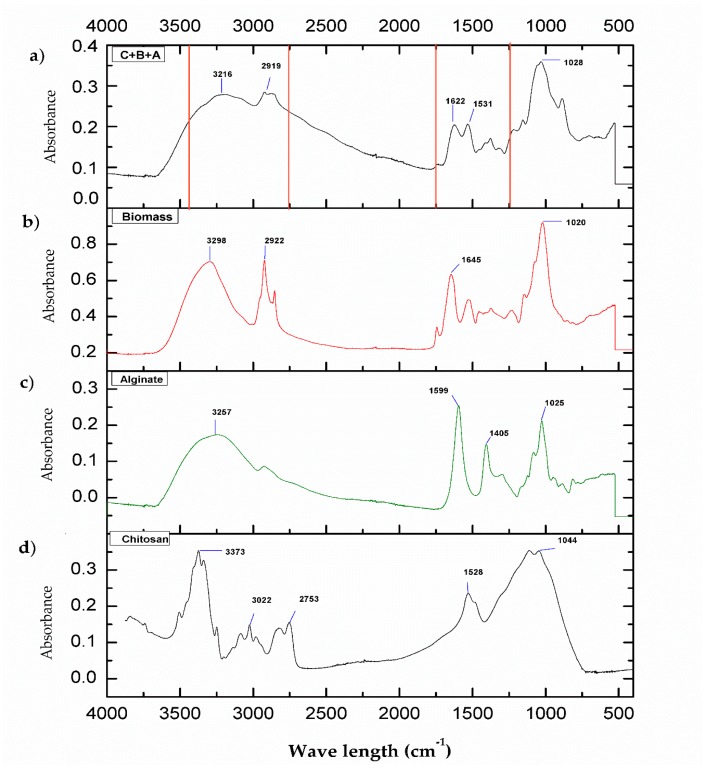
FT-IR of cross-linked chitosan composites with biomass and alginate as follows: (**a**) composite C+B+A; (**b**) fungal biomass; (**c**) sodium alginate, and (**d**) CMMW.

**Figure 3 polymers-11-00222-f003:**
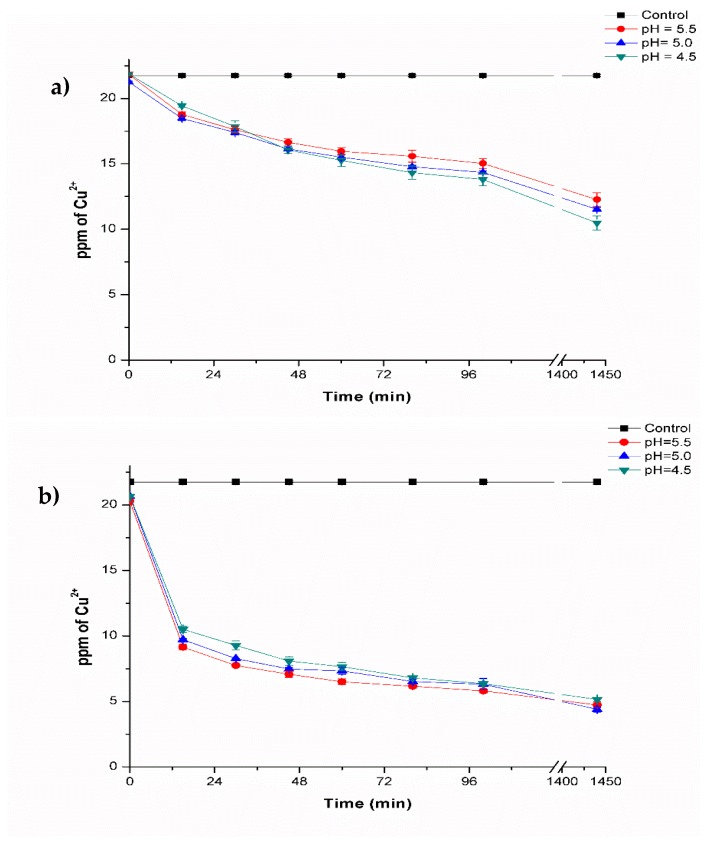
Biosorption of Cu^2+^ at 35 °C at different pH values: (**a**) fresh composite beads, and (**b**) lyophilized composite beads.

**Figure 4 polymers-11-00222-f004:**
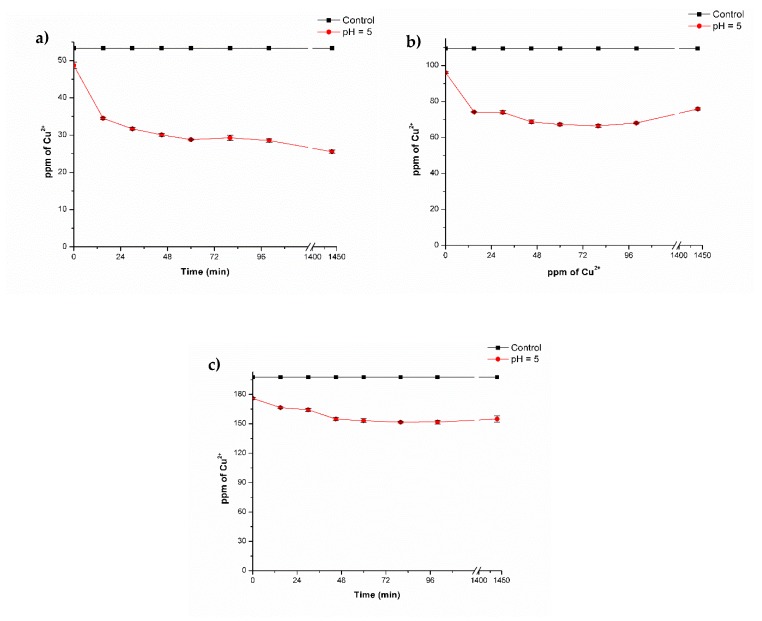
Biosorption of copper with lyophilized composite beads at pH 5 and 35 °C: (**a**) 50 ppm; (**b**) 100 ppm, and (**c**) 200 ppm of Cu^2+^.

**Figure 5 polymers-11-00222-f005:**
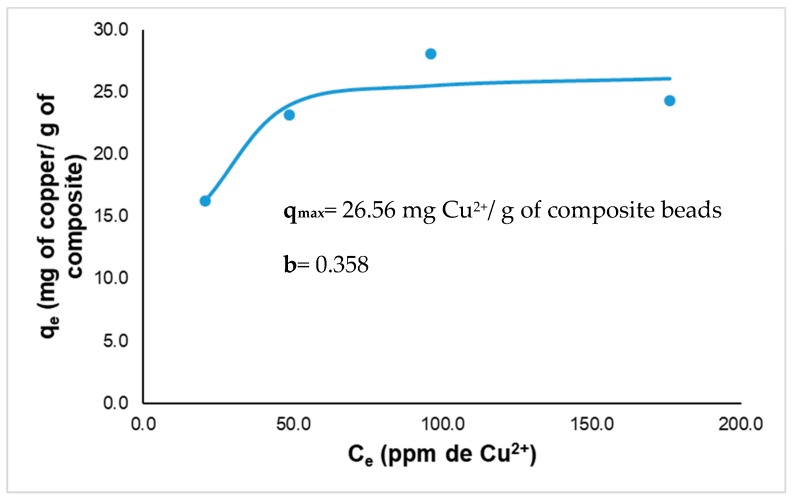
Equilibrium adsorption isotherm using the Langmuir model.

**Figure 6 polymers-11-00222-f006:**
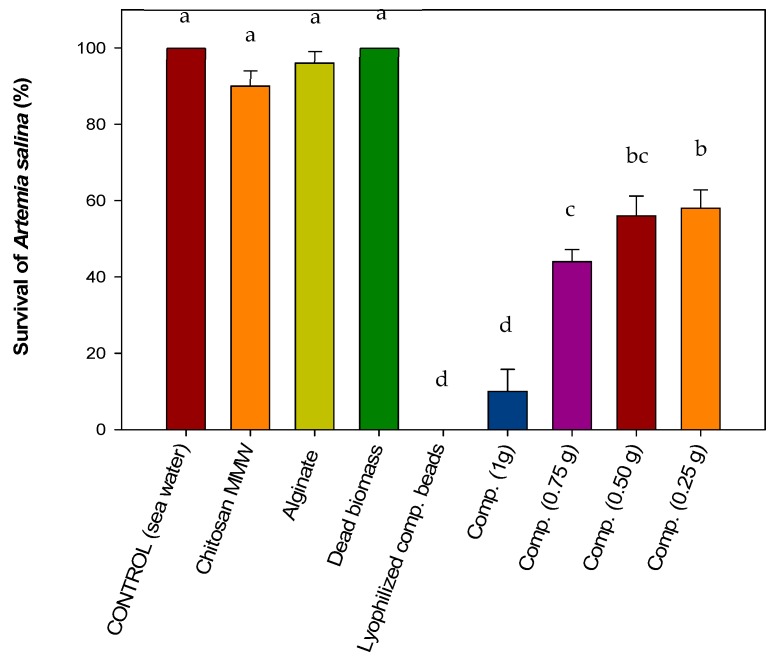
Effect of composite beads on *Artemia salina* nauplii.

**Figure 7 polymers-11-00222-f007:**
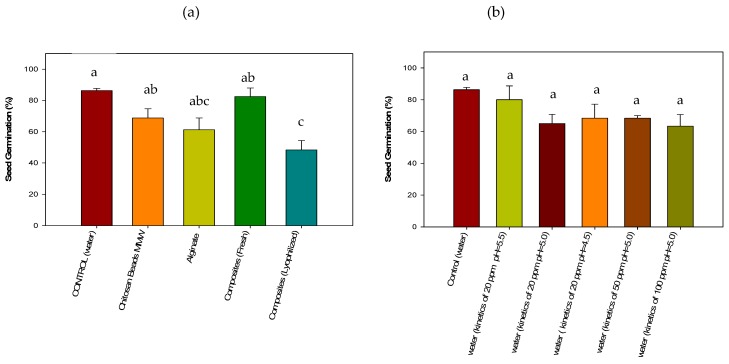
Evaluation of the toxicity of composite beads (**a**) and simulated wastewater after the copper removal treatment (**b**) on the germination of *Lactuca sativa* seeds.

**Figure 8 polymers-11-00222-f008:**
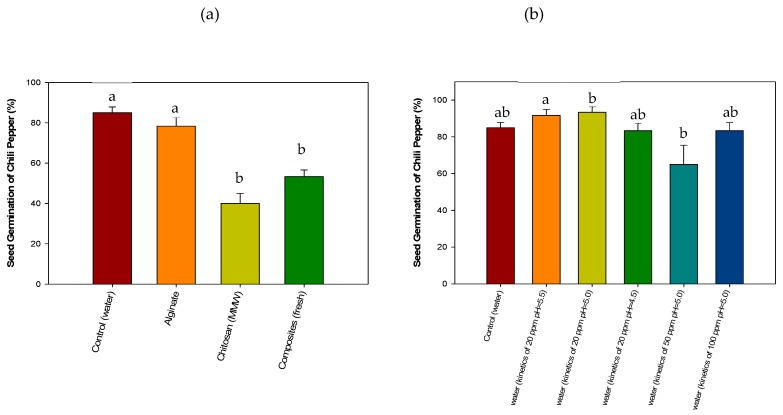
Evaluation of the toxicity of composite beads (**a**) and simulated wastewater after copper removal treatment (**b**) on the germination of *Capsicum annuum* seeds.

**Table 1 polymers-11-00222-t001:** Removal of copper using fresh composite beads and lyophilized composite beads at different pH values, at 35 °C.

	pH	Residual Concentration (ppm of Cu^2+^)	Removal Efficiency (%)
**Control**	5.5	21.765 ± 0.079 ^a^	-
**Fresh Composite**			
F1	5.5	12.26 ± 0.51 ^b^	43.62 ± 2.36
F2	5.0	11.52 ± 0.15 ^bc^	47.04 ± 0.69
F3	4.5	10.48 ± 0.54 ^c^	51.84 ± 2.49
**Lyophilized Composite**			
L1	5.5	4.72 ± 0.20 ^de^	78.29 ± 0.94
L2	5.0	4.39 ± 0.19 ^e^	79.80 ± 0.87
L3	4.5	5.15 ± 0.15 ^d^	76.29 ± 0.71

Values of metal removal (%) and residual copper concentration represents the average of three replicates ± their standard error. Different letters (a, b, c, d, and e) in the super index indicate statistical differences (*P* ≤ 0.05).

**Table 2 polymers-11-00222-t002:** Adsorption of Cu^2+^ into cross-linked chitosan composites with biomass, alginate, and TPP.

Initial Concentration (Co)	Equilibrium Concentration (Ec)	Removal Efficiency (%)	q (mg of Cu^2+^/g of beads)
20.7 ± 0.08	4.39 ± 0.19	79.80 ± 0.87	16.2
48.7 ± 0.82	25.6 ± 0.40	52.07 ± 0.76	23.9
96.1 ± 0.46	66.39 ± 0.87	39.34 ± 0.80	25.5
176.0 ± 0.91	151.61 ± 0.74	23.22 ± 0.37	26.1

Values represents the average of three replicates ± their standard error.

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
