# Peer review of "Toxicological Assessment of Cross-Linked Beads of Chitosan-Alginate and *Aspergillus australensis* Biomass, with Efficiency as Biosorbent for Copper Removal"

_polymers, 2019, doi:10.3390/polym11020222_

Round 1

Reviewer 1 Report

The paper entitled ”Toxicological assessment of cross-linked beads of chitosan-alginate and Aspergillus australensis biomass, with efficiency as biosorbent for copper removal” is interesting and it is well written. Only some minor points should be addressed before publication:

Comments:

-The authors should show differences in biosorption of copper for chitosan before and after cross linking with biomass-alginate and TTP composite beads.

-Lines 193 to 195: Why the composite beads have different size in fresh state (0.2±0.01 cm) and for lyophilized state (0.1±0.01 cm)?.

-Line 262: Why biosorption has been evaluated for pH of 5.5, 5 and 4.5 and why the study has been done in such narrow pH range?.

-Figure 3 and Figure 4: Please, add error bars in the graphs.

-The authors state that maximal adsorption capacity was 26.56 mg Cu2+ per g of composite beads. Is this adsorption capacity enough for a potential application?. Please, compare the result with the literature.

-Lines 512-513: The authors state that water after copper removal should decrease the concentration of copper below toxic levels for plants. Have the authors checked if this has been achieved?.

Minor comments:

Line 151: What does the acronym AAE stands for?. Please, give Information.

Author Response

We are especially thankful with the reviewers and the editor for their efficient work and their important comments to improve this manuscript. All the observations and suggestions were welcomed and considered in the revised manuscript. New references have been included to support the discussion. Revised information is presented in red in the manuscript.

Point 1. The authors should show differences in biosorption of copper for chitosan before and after cross linking with biomass-alginate and TPP composite beads.

Response: Thank you for this important observation, the following discussion has been included in the manuscript: “Biosorption capacity for pure chitosan beads was also evaluated, which were capable of reducing Cu2+ concentration to a value of 1.58 ± 0.94 ppm compared to control (21.67 ± 0.07 ppm). This represents a 92.7 % of metal removal. Pure chitosan beads are able to remove a higher amount of Cu2+ because some active sites are lost during cross-linking process. However, the low mechanical properties of pure chitosan beads after lyophilization are not adequate for biosorption process; these beads can easily break, loose shape, and generate particles with different sizes that are only removed by filtration or centrifugation in comparison to the cross-linked chitosan beads, which can be easily removed by decantation and are resistant to manipulation. In addition, it is reported that pure chitosan beads tend to protonate amino groups present in acidic pH, which causes a partial or total dissolution of pure chitosan in acidic solutions [37]. The cross-linked chitosan beads in this study were resistant to acidic conditions.”

Discussion was included in the manuscript.

Point 2. Lines 193 to 195: why the composite beads have different size in fresh state (0.2 ± 0.01 cm) and for lyophilized state (0.1 ± 0.01 cm)?

Response: In order to respond to reviewer question, the following section has been included in the manuscript: “Fresh (Figure 1a) and lyophilized (Figure 1b) beads show a different size because lyophilized beads lose all water content resulting in a decrease in size [24]. Fresh beads have around 90-95 % of water content, which confirms the hydrophilic nature of the beads [25].”

Information was included in the manuscript.

Point 3. Line 262: Why biosorption has been evaluated for pH of 5.5 and 4.5 and why the study has been done in such narrow pH range?

Response: Biosorption of metals is highly dependent on pH conditions. Cations and protons compete for binding sites, which means that the biosorption of metals such as Cu, Cd, Ni, Co, and Zn is reduced at low pH values (below 3.0) [39]. In the present study, pH values within 4.5 – 5.5 ranges were used based on a previous study on biosorption processes, in which Aspergillus australensis biomass (data not shown) performed better for copper removal within this pH range. Also, at a pH above 5.8 – 6, a precipitation of the metal can occur in the form of Cu(OH)2 hydroxides, therefore it is not recommendable to carry out biosorption processes at alkaline conditions. This information was included in the corrected manuscript.

Point 4. Figure 3 and Figure 4: Please, add error bars in graphs.

Response: Graphs shown in figures 3 and 4 were modified in order to appreciate the error bars. Thank you.

Point 5. The authors state that maximal adsorption capacity was 26.56 mg Cu2+ per g of composite beads. Is this adsorption capacity enough for a potential application? Please, compare the result with the literature.

Response: We are really grateful for this important observation. More discussion and new references were included in the manuscript.

“Comparing with other adsorbent materials, Mokhter et al. [42] used modified silica-chitosan to remove 68 mg g-1 of Cu2+. Shim et al. [43] and Monser and Adhoum [44] used activated carbon and they achieved to removed 9 - 38 mg g-1 copper; Babel et al. [45] used clay adsorbents to remove around 1 - 25 mg g-1 copper as well. Comparing the values of copper removal reported in the literature, it is considered that the composites of chitosan cross-linked with fungal biomass, alginate, and TPP have a good biosorption capacity for metals such as copper, and they represent a feasible and low cost option for biosorbent process application.”

Point 6. Lines 512-513: The authors state that water after copper removal should decrease the concentration of copper below toxic levels for plants. Have the authors checked if this has been achieved?

Response: Thank you very much for this important observation. In this study, copper concentration was reduced to a minimum of 4.72 ± 0.2 – 4.39 ± 0.19 ppm of Cu2+. As previously mentioned, the remaining concentration was not below toxic levels for seed germination and adequate growth of lettuce and chili pepper seeds. However, copper concentration can further be decreased by implementing 2-way kinetics strategies to increase copper removal; therefore, more studies are needed to optimize the biosorption process presented in this study. This information was included in the manuscript.

Point 7. Minor comments:

Line 151: What does the acronym AAE stands for? Please, give information.

Response: The acronym AAE stands for Atomic Adsorption Spectroscopy. Correction was made and AAE was changed for AAS.

Reviewer 2 Report

The paper is well-written, well-documented with a good literature database and the results are reliable and nicely discussed. For these reasons, I recommend its publication, notwithstanding some minor corrections listed below.

- Maybe the authors must not ignore the following reference, which present good copper retention capacity by modified silica particles: Mokhter et al, Colloids Interfaces, 2018, 2(2) 19.

- Specify the method for the acquisition of the FT-IR spectra (transmission, diffusion, ATR?)

- For the biosorption of copper, please specify that the aliquots of 10 mL were taken under shaking and by assuring the solid/liquid ratio were not changed by this operation. I guess it is the case, but it is better to specify it because the adsorption features are greatly dependent on the solid content in the solution.

- In the Langmuir model, the limit of adsorption is interpreted as a complete covering of the adsorbent surface as written by the authors. But the authors should precise that this isotherm model assumes the formation of an adsorbed solute monolayer on a uniform surface with a finite number of adsorption sites. This means that when a site is filled, no further sorption can take place at that site. Therefore, the surface will reach a saturation point where the maximum adsorption of the surface will be achieved. Moreover, the Langmuir constant (b) is related to the energy of adsorption. Its value may be interpreted as follows: b<0: unfavorable adsorption; b=0: linear adsorption with no specific affinity (Freundlich-type); b>0: favorable adsorption (the more b, the better affinity).

- I am not sure I would write that at pH 4.5, the lyophilized beads exhibit slightly less copper removal than at the two other pHs. I think that for the lyophilized beads, the removal is the same at the three pH within the precision of the measurements. A more significant effect can be seen with the fresh beads, with an increase of the copper removal when the pH decreases. In addition to the amine groups (which protonated and thus positively charged for pH between 4.5 and 5.5), the other relevant groups to consider are the carboxyl groups whose pKa is between 4.5 and 5.0.

- Pages 9-10: I understand what the authors mean but it is not properly written: the copper removal does not decrease by increasing copper concentration as written at the very beginning of page 10. But rather, the amount of copper that the composite beads could adsorb was higher with increasing copper concentration, as written just after. These two quotations mean exactly the reverse. Of course, what is called the removal efficiency (%) decreases by increasing copper concentration, but because the copper concentration increases more than the copper removal, which increases as well.

- Page 13: I would write “the different waters after treatment are equally harmful for seed germination” instead of “water after treatment is not harmful for seed germination”. Because, one can notice a decrease in the seed germination with the treated waters compared to the control (except the first one at 20 ppm pH 5.5 which should be discussed apart).

- Copper can affect the biosorption of nutrients and other important physiological processes when present in excess [48]: please specify “in excess” by giving values or at least orders of magnitude.

Typos:

Line 20: considered as an alternative

Line 51: considered as an alternative

Line 79: another instead of other

Line 131: spectra

Line 445: attributed

Figure 5: use the same legends as in the Langmuir equation i.e., qe and Ce instead of q and C0.

Author Response

We are especially thankful with the reviewers and the editor for their efficient work and their important comments to improve this manuscript. All the observations and suggestions were welcomed and considered in the revised manuscript. New references have been included to support the discussion. Revised information is presented in red in the manuscript.

Point 1. Maybe the authors must not ignore the following reference, which present good copper retention capacity by modified silica particles: Mokhter et al. Colloids Interfaces, 2018, (2) 19.

Response: Thank you for the comment. The reference was revised and included for comparison and discussion.

Point 2. Specify the method for the acquisition of the FT-IR spectra (transmission, diffusion, ATR?).

Response: The method used was ATR (Attenuated Total Reflectance). Information was inserted in the manuscript.

Point 3. For the biosorption of copper, please specify that the aliquots of 10 mL were taken under shaking and by assuring the solid/liquid ration were not changed by this operation. I guess it is the case, but it is better to specify it because the adsorption features are greatly dependent on the solid content in the solution.

Response: We are really grateful for this important observation. The information was added to the text where the method is described: “The Erlenmeyer flasks were placed in an orbital shaker at 120 rpm and incubated at 35°C. Samples of 10 mL were taken during shaking assuring that the solid/liquid ratio were not changed in the process”.

Point 4. In the Langmuir model, the limit of adsorption is interpreted as a complete covering of the adsorbent surface as written by the authors. But the authors should precise that this isotherm model assumes the formation of an adsorbed solute monolayer on a uniform surface with a finite number of adsorption sites. This means that when a site is filled, no further sorption can take place at the site. Therefore, the surface will reach a saturation point where the maximum adsorption of the surface will be achieved. Moreover, the Langmuir constant (b) is related to the energy of adsorption. Its value may be interpreted as follows: b<0: unfavorable adsorption; b=0: linear adsorption with no specific affinity (Freundlich-type); b>0: favorable adsorption (the more b, the better affinity).

Response: Thank you for the comment. The next paragraph was added: “Langmuir isotherm model assumes the formation of an adsorbed solute monolayer on a uniform surface with a finite number of adsorption sites. This means that when a site is filled, no further sorption can take place at that site; therefore, the surface will reach a saturation point where the maximum adsorption of the surface will be achieved. In addition, the Langmuir constant (b) is related to the adsorption energy that may be interpreted as follows: b < 0: unfavorable adsorption; b = 0: linear adsorption with no specific affinity; b > 0: favorable adsorption. Higher values of b suggest better affinity.”

Point 5. I am not sure I would write that at pH 4.5, the lyophilized beads exhibit slightly less copper removal than at the two other pH’s. I think that for the lyophilized beads, the removal is the same at the three pH within the precision of the measurements. A more significant effect can be seen with the fresh beads, with an increase of the copper removal when the pH decreases. In addition to the amine groups (which protonated and thus positively charged for pH between 4.5 and 5.5) the other relevant groups to consider are the carboxyl groups whose pKa is between 4.5 and 5.0.

Response: The information was re-written based on suggestions. “The results obtained with lyophilized composite beads are presented in Figure 3b. The copper concentration also decreased in the pH evaluated where, at pH 4.5, 5.0 and 5.5, copper removal was observed with values of 5.15 ± 0.15 ppm, 4.39 ± 0.19 and 4.72 ± 0.20 ppm of Cu2+, respectively. Removal efficiency was found to be 76.2%, 79.8 % and 78.2% (Table 1), where no differences were found among the treatments. The amount of metal removed by lyophilized composite beads was around 17.11 mg of Cu2+ per g of composite.”

Point 6. Pages 9-10: I understand what the authors mean but it is not properly written; the copper removal does not decrease by increasing copper concentration as written at the very beginning of page 10. But rather, the amount of copper that the composite beads could adsorb was higher with increasing copper concentration, as written just after. These two quotations mean exactly the reverse. Of course, what is called the removal efficiency (%) decreases by increasing copper concentration, but because the copper concentration increases more than the copper removal, which increases as well.

Response: The information was re-written. “With the results obtained in this kinetic, the Langmuir model was applied (Figure 5, Table 2) and analysis showed that the amount of copper that the composite beads could adsorb was higher with increasing copper concentration. Notwithstanding a decrease in removal efficiency (%) was observed for biosorption process, but this could be due to the increase in Cu2+ concentration; nonetheless, a higher amount of metal was adsorbed for each concentration evaluated.” Thank you.

Point 7. Page 13: I would write “the different waters after treatment are equally harmful for seed germination” instead of “water after treatment is not harmful for seed germination”. Because, one can notice a decrease in seed germination with the treated waters compared to the control (except the first one at 20 ppm PH 5.5 which should be discussed apart).

Response: Corrections were made. “In Figure 7b, the effect of the residual water from biosorption kinetics are presented and, as it can be observed, no statistical differences (p > 0.05) were found among the treatments, indicating that the different waters after treatment are potentially equally harmful for seed germination. The treatment that was less affected was the residual water of the kinetics with 20 ppm of Cu2+ and pH 5.5, were seed germination was similar to control.”

Point 8. Copper can affect the biosorption of nutrients and other important physiological processes when present in excess [48]: please specify “in excess” by giving values or at least orders of magnitude.

Response: Information of copper values and new references were included. “Copper can affect the biosorption of nutrients and other important physiological processes when present in excess, for example, at concentrations of 20 – 30 µg g-1 dry weight [56,57].” Thank you.

Point 9. Typos:

Line 20: considered as an alternative

Line 51: considered as an alternative

Line 79: another instead of other

Line 131: spectra

Line 445: attributed

Response: Corrections in lines 20, 51, 79, 131 and 445 were made.

Point 10. Figure 5. Use the same legends as in the Langmuir equation i.e. qe and Ce instead of q and C0.

Response: Corrections were made in figure 5. Thank you.
